# New Paradigms for Cytoreductive Nephrectomy

**DOI:** 10.3390/cancers14112660

**Published:** 2022-05-27

**Authors:** Benjamin J. Lichtbroun, Arnav Srivastava, Sai Krishnaraya Doppalapudi, Kevin Chua, Eric A. Singer

**Affiliations:** Section of Urologic Oncology, Rutgers Cancer Institute of New Jersey, Rutgers Robert Wood Johnson Medical School, 195 Little Albany Street Room 4563, New Brunswick, NJ 08901, USA; blichtbr@rwjms.rutgers.edu (B.J.L.); srivasar@rwjms.rutgers.edu (A.S.); sd839@rwjms.rutgers.edu (S.K.D.); kc1133@rwjms.rutgers.edu (K.C.)

**Keywords:** cytoreductive nephrectomy, immune-oncologic agents, metastatic renal cell carcinoma, immunotherapy, renal cell carcinoma, cytoreductive surgery

## Abstract

**Simple Summary:**

Cytoreductive surgery (CS) is performed to remove the primary tumor in the setting of metastatic disease. In metastatic renal cell carcinoma (mRCC), the role of cytoreductive nephrectomy (CN) in the treatment paradigm has evolved, adjusting to new changes in systemic therapy agents. In particular, immunotherapeutic agents, which utilize the body’s own immune system to attack cancerous cells, have improved over the past decade. Newer immunotherapy agents offer more effective treatments in mRCC, with the goal of more tolerable side effect profiles. However, now urologic and medical oncologists must reframe the role of CN in the context of these new systemic therapies. This review will discuss the current data on this topic as well as the historical context in which it is being studied.

**Abstract:**

The role of CN in the treatment of metastatic renal cell carcinoma (mRCC) has been studied over the course of the past few decades. With the advent of immuno-oncologic (IO) agents, there has been a paradigm shift in the treatment of RCC. Within this new era of cancer care, the role of CN is unclear. There are several studies currently underway that aim to assess the role of CN in combination with these therapies. We reviewed articles examining CN, both historically and in the modern immunotherapy era. While immune-oncologic agents are relatively new and large clinical trials have yet to be completed, data thus far is promising that CN may provide clinical benefit. Multiple ongoing trials may clarify the role of CN in this new era of cancer care.

## 1. Introduction

Renal cell carcinoma (RCC) is the sixth most commonly diagnosed cancer in men and ninth in women, accounting for 5% and 3% of all cancers detected in these populations, respectively [1]. Many patients with RCC present with advanced disease, with as many as 17% of patients harboring distant metastasis at the time of diagnosis [2]. Interestingly, 20–40% of patients who present with localized disease will ultimately develop distant metastasis [3]. Cytoreductive surgeries, which remove the primary tumor in the presence of metastatic disease, have been proposed as a possible option in these patients in order to reduce the tumor burden with a goal to improve quality of life and survival. Cytoreductive surgeries have shown clinical benefit with other malignancies such as breast, ovarian, and various abdominal malignancies, and possibly even with other genitourinary malignancies [4,5,6]. The treatment of metastatic RCC (mRCC) has drastically changed over the course of the past few decades [7,8,9]. With the introduction of the new immune-oncologic treatment options in mRCC, the role of CN is again unclear. In this review, we discuss the contemporary role of CN in patients with metastatic RCC and how these surgeries fit in to the current treatment landscape for mRCC. 

## 2. Materials and Methods

Using Pubmed, Google Scholar, and Wiley Online Library, we performed a review of articles between 2000 and 2022. Search terms included a combination of terms including “cytoreductive nephrectomy”, “immunotherapy”, and “metastatic renal cell carcinoma”. Articles included were original articles published in English. Unpublished works, works not in English, and news articles were not included. Information on clinical trials was collected from clinicaltrials.gov, which was accessed in March 2022. 

## 3. Results

### 3.1. The Cytokine Period

During the cytokine era in the early 2000s, the SWOG group performed a prospective, randomized trial to assess whether a nephrectomy offered a survival benefit in patients with mRCC. Patients with mRCC were offered a nephrectomy followed by interferon alpha-2b compared with others who received interferon alpha-2b alone. In this study, there was a median survival benefit of 11.1 months vs. 8.1 months (*p* = 0.05) in the group that received surgery [10]. During that time, the EORTC group also published a similar study looking at nephrectomy plus interferon therapy compared with interferon therapy alone in patients with mRCC. In their study, they found a PFS benefit of 5 months vs. 3 months in the surgery group (HR 0.6, 95% CI 0.36–0.97, *p* = 0.04) and an OS benefit of 17 months vs. 7 months in the surgery group (HR 0.54, 95% CI 0.31–0.94, *p* = 0.03) [11]. Given the similarities in these prospective randomized trials, the groups later published a combined analysis of their data. When assessed in this way, using an intention-to-treat analysis, the overall median survival was 13.6 months in the CN group and 7.8 months in the interferon-alone group (HR 0.69, 95% CI 0.55–0.87, *p* = 0.002). There was a 31% decrease in risk of death in the nephrectomy group. There was a 51.9% 1-year survival rate with the nephrectomy group compared to 37.1% in the interferon-alone group. 

The combined analysis provided a more accurate assessment of the treatment difference associated with CN. For instance, in the SWOG study alone, when controlling for performance status, there was no significant survival difference. With this combined analysis, there was now a significant survival difference when controlling for performance status. Given the larger sample size, prognostic factors were also better assessed. Performance status was shown to have prognostic importance, while the site of metastasis and disease measurability did not [12]. The SWOG group then published the long-term data from their initial study with median follow-up of 9 years, which continued to show long-term overall survival benefits with cytoreductive surgery. This benefit was seen in all predefined patient strata, including performance status, the presence or absence of lung metastasis, and the presence or absence of measurable disease [13]. While the landscape of treatment for mRCC has changed drastically since these publications, they laid the framework for future studies aimed at investigating the role of CN. 

### 3.2. Targeted Therapies

In the years following the publication of these aforementioned landmark studies, there were improvements in our understanding of the molecular basis of RCC. At that time, there was a transition to the use of targeted therapies such as VEGF inhibitors, mTOR inhibitors, and tyrosine kinase inhibitors (TKIs). In 2010, Choueiri et al. performed a retrospective review looking at 314 patients with mRCC who either received VEGF targeted therapy alone or VEGF targeted therapy with CN [14]. The specific therapies used were sorafenib, sunitinib, or bevacizumab. In this study, there was a median OS benefit of 19.8 months vs. 9.4 months in the surgery group compared to the VEGF-alone group (HR 0.44, 95% CI 0.32–0.59, *p* < 0.01) [14]. The National Cancer Database was then used in order to assess the survival benefit of CN in patients treated with targeted therapy. In the 15,390 patients treated with targeted therapy at that time, 5374 (35%) underwent CN between 2006–2013. The OS was 17.1 months vs. 7.7 months in the cytoreductive nephrectomy group compared to targeted therapy alone (*p* < 0.001). There were multiple patient factors and socioeconomic factors associated with receiving a CN, including being younger, being treated at an academic center, being privately insured, having a lower tumor stage, and having N0 [15]. A meta-analysis was also performed at that time looking at the role of CNs in patients treated with targeted therapies, which also showed an overall survival benefit with CN (HR 0.46, 95% CI 0.32–0.64, *p* < 0.01) [16].

Up until 2017, there were numerous studies supporting nephrectomies performed in the cytoreductive setting. In 2018, the results of the CARMENA trial called into question the utility of CN in the targeted therapy era. The CARMENA trial was a prospective, randomized, phase III trial. It was a non-inferiority design that compared sunitinib alone vs. sunitinib after CN in patients with mRCC. In this study, only intermediate-risk and poor-risk patients were included. With regard to OS, the sunitinib-alone group was found to be non-inferior to the CN group—18.4 months vs. 13.9 months (HR 0.89, 95% CI 0.71–1.10). One critique was that the study was underpowered as they were only able to accrue 450 of a planned 576 subjects. The study was also slow to accrue, in that it took 8 years to accrue the necessary patients across 79 centers. This accounts for 0.7 subjects per center per year, which points out that the study sites were either low-volume centers or there was a lack of equipoise, and not all providers recommended trial participation to all potential participants. Patient selection was also a major issue in that the study had mostly poor-risk patients with a high burden of disease. Furthermore, 57% of these patients had high-risk disease according to the MSKCC/Motzer score. In addition, 72% of the patients in this trial had non-lung metastasis with a median tumor burden of 14.2 cm, making this population particularly high-risk. There was also significant cross-over between the two groups with 15% of the patients in the nephrectomy group not receiving sunitinib and 17% of the patients in the sunitinib-alone group undergoing a nephrectomy. In addition, the 18.4-month OS reported in the sunitinib group was lower than other previously published reports, again raising questions about patient selection and generalizability. While the sunitinib-alone arm did show non-inferiority, the previously mentioned critiques largely limited its broad applicability [17].

Shortly after the publication of the CARMENA trial, the results of the SURTIME trial were published. In this trial, the authors compared immediate vs. deferred CN in patients with mRCC receiving sunitinib therapy [18]. Initially, the primary end point of the study was set to be PFS with an initial sample size of 458 patients. Due to low accrual at 3 years, the independent data monitoring committee endorsed reporting a 28-week progression-free rate and decreased the sample size to 98 patients. OS, adverse events, and post-operative progression were secondary end points. The 28-week progression-free rate was 42% in the immediate CN and 43% in the deferred CN arm. The intention-to-treat OS hazard ratio of immediate vs. deferred CN was 0.57 with a median OS of 32.4 months in the deferred arm vs. 15.0 months in the immediate arm (95% CI 9.3–29.5 months, *p* = 0.03).

Much like the CARMENA trial, this study also had certain significant limitations. For one, there was low accrual. In addition, the primary end point with 28-week progression-free survival required complex timing in order to appropriately ascertain these data. Notably, the superiority of the combination of nivoulumab and ipilimumab over sunitinib, in terms of survival and quality of life, changed the first-line therapy for patients with intermediate and poor-risk mRCC, limiting the applicability of the results of the SURTIME and CARMENA trials [18]. The authors of the CARMENA trial did later perform a post-hoc analysis of overall survival in patients who had a secondary nephrectomy. A total of 40 patients (18%) in the sunitinib-alone group ultimately underwent secondary CN. Of those, 31% resumed sunitinib therapy after surgery. The patients who underwent secondary CN had a significantly longer OS than those who did not undergo surgery at all—48.5 months vs. 15.7 months, respectively (HR 0.34, 95% CI 0.22–0.54) [19].

The SURTIME group did later assess their data with regard to surgical safety. When comparing the immediate and deferred CN groups, the rates of all adverse events were essentially the same—52% vs. 53%, respectively [20]. These data show that while CN is of course a morbid procedure, the use of TKIs in either setting did not increase the rates of adverse events [20]. Following this publication, an analysis on the Registry for Metastatic Renal Cell Carcinoma registry was performed in order to measure the rates and predictors of perioperative complications [21]. Data from 736 mRCC patients undergoing CN at 14 institutions were retrospectively studied. Logistical regression analysis was used in order to identify predictors of intraoperative complications, post-operative complications, as well as 30-day readmission rates. Intraoperative complications were seen in 10.9% of patients. Interestingly, 29.5% of patients encountered a post-operative complication of any grade, with 6.1% encountering high grade complications. The 30-day readmission rate was 11.5% overall. The CN case load at each center was strongly inversely correlated with high-grade post-operative morbidity, highlighting the importance of centralizing these surgically complex procedures [21].

### 3.3. Optimizing Patient Selection

In patients undergoing CN, Akimi et al. aimed to determine the association between certain modifiable International Metastatic Renal Cell Carcinoma Database Consortium (IMDC) risk factors and oncologic outcomes. Furthermore, 245 patients were treated at a single institution between 2009 and 2019 [22]. The primary variable of interest was the type and number of IMDC risk factors such as anemia, hypercalcemia, neutrophilia, thrombocytosis, and reduced Karnofsky performance status at the time of initial clinical evaluation. The final IMDC risk factor, “less than 1 year from diagnosis to systemic therapy”, was not assessed in the treatment-naïve cohort as it was calculated at the time of commencing therapy. In the patients who underwent CN, IMDC-modifiable risk factors were assessed at 6 weeks and 6 months after surgery with the Karnofsky performance status calculated each time. Radiographic imaging was used to assess disease burden each time. Sites were grouped broadly into viscera, bones, nodes, or other sites. They were also grouped as either unifocal (one organ) or multifocal (multiple organs). In all time points—pre-operatively, 6 weeks post-op, and 6 months post-op—higher numbers of IMDC risk factors were all associated with adverse overall survival (HR 1.41, 95% CI 1.19–1.68, *p* < 0.001; HR 1.55, 95% CI 1.25–1.92, *p* < 0.001; HR 2.43, 95% CI 1.85–3.21, *p* < 0.001). In patients who had a decrease in IMDC risk factors at 6 weeks, they were shown to have improved overall survival (HR 0.64, 95% CI 0.46–0.89, *p* = 0.007). When assessing only high-risk patients with 2 or more IMDC risk factors, patients who were able to reduce the number of IMDC risk factors to one or fewer saw an increase in overall survival at 6 weeks (*p* = 0.36) and 6 months (*p* < 0.001) [22]. A recent retrospective multicenter study aimed to evaluate the survival benefit of upfront CN in mRCC, stratified by IMDC risk factors. Charts were reviewed for patients who received upfront CN, deferred CN, and systemic therapy alone over an 11-year span. Of the 259 patients who met the inclusion criteria, 107 were classified as being in the upfront CN group. After inverse probability of treatment weighting, upfront CN was found to have a survival benefit in patients with IMDC intermediate- and poor-risk disease [23].

McIntosh et al. recently performed a retrospective study aimed at identifying risk factors associated with patients less likely to benefit from CN in the targeted therapy era [24]. In this study, 608 patients underwent CN between 2005 and 2017. Patients were retrospectively assessed based off of pre-operative and peri-operative clinical data, laboratory data, and pathologic data. Clinical factors that were associated with a decrease in overall survival were systemic symptoms at diagnosis (HR 1.24, 95% CI 1.01–1.52), retroperitoneal lymphadenopathy (HR 1.39, 95% CI 1.12–1.71), supradiaphragmatic lymphadenopathy (HR 1.41, 95% CI 1.07–1.86), bone metastasis (HR 1.42, 95% CI 1.14–1.77), and clinical T4 disease (HR 1.87, 95% CI 1.18–2.95). Pre-operative laboratory data associated with decreased OS were hemoglobin less than the lower limit of normal (HR 1.33, 95% CI 1.08–1.66), serum albumin less than the lower limit of normal (HR 1.41, 95% CI 1.07–1.85), serum LDH greater than the upper limit of normal (HR 1.55, 95% CI 1.23–1.96), and a neutrophil/lymphocyte ratio greater than or equal to four (HR 1.46, 95% CI 1.14–1.86). Based off of these nine risk factors that were determined to decrease OS, patients were stratified based off of number of risk factors present—low-risk (0–1 risk factors), intermediate-risk (2–3 risk factors), and high-risk (>3 risk factors). The risk of death was found to be proportional to number of pre-operative risk factors present. Median OS was found to be 58.9 in the low-risk group (95% CI 44.3–66.6 months), 30.6 months in the intermediate-risk group (95% CI 27.0–35.0 months), and 19.2 (95% CI 13.9–22.6 months) in the high-risk group. Adverse features at final pathology were also more common in the high-risk group. Pathologic T and N stages, tumor size, positive margin rate, and non-clear cell histology were associated with being in the high-risk group (all *p* < 0.01). In terms of perioperative variables, being in the high-risk group was associated with an increased estimated blood loss (*p* = 0.06), length of hospitalization (*p* = 0.008), postoperative complication rate (*p* = 0.009), and readmission rate (*p* < 0.001) [24].

### 3.4. Immune Checkpoint Inhibitors—What We Know

Immune checkpoint inhibitors have ushered in a contemporary era of immunotherapy in mRCC. First with the CheckMate 025 trial and subsequently with the CheckMate 214, KEYNOTE-426, CLEAR, IMmotion151, and JAVELIN Renal 101 trials, immunotherapies have become a staple in the management of mRCC. In the metastatic setting, immune checkpoint inhibitor combination therapies have led to primary tumor shrinkage [25]. With the advent of immune checkpoint inhibitors, the role for cytoreductive surgery remains unknown (Figure 1). A recent NCDB study reviewed 391 surgical candidates diagnosed with metastatic clear cell RCC. Patients were treated with either contemporary immunotherapy alone or combined with CN. No other systemic therapies were used in this cohort. Patients who underwent CN and received immunotherapy had a significantly better OS than those who received immunotherapy alone (HR 0.23, 95% CI 0.15–0.37, *p* < 0.001) [26].

To highlight the superiority of newer immune checkpoint inhibitors over the previous first-line therapy, sunitinib, in advanced renal cell carcinoma, a recent Phase 3, open-label, randomized trial compared sunitinib alone against nivolumab with cabozantinib. This study found that in the non-cytoreductive setting, the combination of nivolumab and cabozantinib offered an increase in overall survival, progression-free survival, and likelihood of response [27].

**Figure 1 cancers-14-02660-f001:**
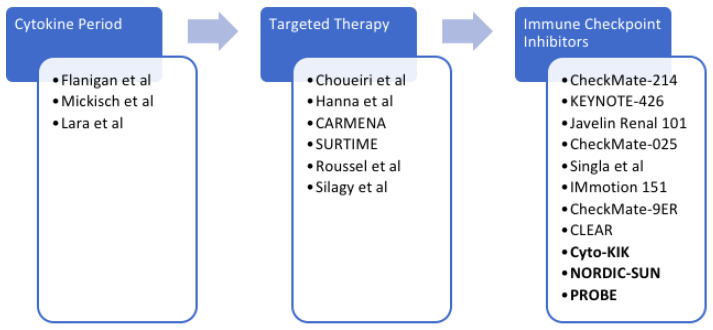
Depiction of the different eras of systemic therapy and the landmark papers evaluating the role of CN during these periods [10,11,13,14,15,17,18,21,22,25,28].

### 3.5. Immune Checkpoint Inhibitors—Ongoing Trials

In the contemporary immunotherapy era, the role of CN has yet to be clearly elucidated by publications with high levels of evidence. Currently, there are multiple trials looking at this exact clinical question (Table 1) [9]. With the superiority of nivolumab and cabozantinib over sunitinib in the advanced renal cell carcinoma population, researchers are aiming to determine the complete response rate in patients receiving neoadjuvant nivolumab and cabozantinib followed by nephrectomy and subsequent systemic therapy in the Cyto-KIK trial [28]. Within this study, patients will also have renal mass biopsies prior to beginning treatment in order to determine biomarkers of response in this population through analysis of RNA sequencing. Comparing these samples to the nephrectomy specimens may help to clarify the mechanism of action in responding patients and the mechanism of resistance in non-responders [28].

Other ongoing studies, including the NORDIC-SUN and PROBE trials, are comparing patients who are receiving immune checkpoint inhibitors alone or in combination compared to patients receiving those medications in addition to cytoreductive surgery [28]. In the PROBE trial, researchers are evaluating if CN offers a survival benefit in patients who had an objective response or stable disease at metastatic sites after receiving any of the multiple first-line options for systemic therapy. This study includes patients with histologically proven clear cell RCC or non-clear cell RCC. Patients will be evaluated after 12 weeks of systemic immune checkpoint inhibition to assess their response to treatment and whether there is benefit in performing a CN.

The NORDIC SUN trial is evaluating the role of deferred CN in patients who have at least three IMDC high-risk features and are receiving combination nivolumab and ipilimumab [25]. The rationale behind the deferred CN is that it still provides the benefit of surgery while not delaying systemic immune checkpoint inhibition. It also spares surgery in patients with progressive tumors. The primary end point in both the NORDIC SUN and PROBE trials is OS.

## 4. Discussion

Cytoreductive surgery has been well studied and shows clinical benefit in the treatment of many non-genitourinary malignancies. The role of cytoreductive surgery in the setting of metastatic renal cell carcinoma has been studied at length with its role changing as the treatment paradigms shift. Cytoreductive surgery was first shown to be beneficial when used in conjunction with interferon therapy. While we no longer routinely use interferon therapy in the management of metastatic RCC, it proved that there may be a role for surgery in these patients. After the molecular basis of renal cell carcinoma was better understood and the targeted therapy era began, we again learned of the clinical utility of cytoreductive nephrectomy for these patients.

The publication of the CARMENA trial and SURTIME trials, for a time, challenged the role of cytoreductive nephrectomies for patients receiving single-agent frontline VEGF inhibition. The limitations of these studies highlight the importance of appropriate patient selection and prospective publications with high levels of evidence in the contemporary treatment era. The SURTIME data did however highlight that the addition of sunitinib did not increase the morbidity of cytoreductive nephrectomies. Shortly after the SURTIME data, the analysis on the Registry for Metastatic Renal Cell Carcinoma registry showed that there was an inverse relationship between case volumes and morbidity, highlighting the importance of doing these highly complex procedures at institutions with experienced surgical teams and the appropriate resources to care for them both in the peri-operative setting as well as the post-operative setting. In terms of patient selection, patients with less IMDC risk factors were more likely to have favorable outcomes and if patients could decrease the amount of IMDC risk factors during treatment then they were more likely to have a favorable outcome. Finally, in the modern era of immune checkpoint inhibitors, the role of cytoreductive surgery remains largely uncertain. A recent NCDB study showed that it likely does have clinical benefit. Currently, there are multiple randomized trials investigating the role of cytoreductive nephrectomies with immune checkpoint inhibition.

Within the exciting landscape that is the contemporary era of cancer care, there remain myriad new opportunities for cytoreductive surgical trials with systemic therapies. The question of whether these therapies should be given in the neoadjuvant setting, adjuvant setting, or both is yet to be clearly elucidated in the cytoreductive setting. The optimal combination and safety of these medications in conjunction with cytoreductive surgery is actively being studied, but results have yet to be published. Manipulation of the gut microbiome is also being studied with promising results as a possible mechanism to enhance the efficacy of immune checkpoint inhibitors in mRCC [29]. The role of prehabilitation in conjunction with enhanced recovery after surgery (ERAS) pathways have largely shown clinical benefit for many genitourinary surgeries—their role in the cytoreductive nephrectomy setting has yet to be described in the immune-oncologic era [30,31,32]. With the combination of toxicities from surgery and systemic therapy, the impact of the early integration of palliative care into the management of mRCC has yet to be studied in this particular population. Many authors argue that palliative care should be introduced at the time of diagnosis or parallel to curative treatment in cases of life-limiting diseases [33]. Clinical trial support/participation will be critical to help answer these important questions surrounding the role of CN in mRCC.

## 5. Conclusions

The role of cytoreductive nephrectomies has evolved over the past few decades. As we await the results of the Cyto-KIK, NORDIC-SUN, and PROBE trials to help better elucidate the role of CN in the current landscape of immune-oncologic care, the current data indicate that there is a benefit in performing CN in select patients with mRCC.

## Figures and Tables

**Table 1 cancers-14-02660-t001:** Ongoing trials looking at cytoreductive nephrectomy in the immune checkpoint inhibitor era.

Study Name	Trial Number	Status	Primary Endpoint	Intervention
Deferred Cytoreductive Nephrectomy in Synchronous Metastatic Renal Cell Carcinoma: The NORDIC-SUN- Trial	NCT03977571	Recruiting	Overall survival	Nivolumab, ipilimumab, cytoreductive nephrectomy
Comparing the Outcome of Immunotherapy-Based Drug Combination Therapy with or Without Surgery to Remove the Kidney in Metastatic Kidney Cancer, the PROBE trial	NCT04510597	Recruiting	Overall survival	Cytoreductive nephrectomy, active comparator
CYTO-reductive Surgery in Kidney Cancer Plus Immunotherapy and Targeted Kinase Inhibition (CYTO-KIK)	NCT04322955	Recruiting	Complete response rate	Cabozantinib, nivolumab, cytoreductive nephrectomy
Nivolumab With or Without Bevacizumab or Ipilimumab Before Surgery in Treating Patients with Metastatic Kidney Cancer That Can Be Removed by Surgery	NCT02210117	Active, not recruiting	Adverse events	Bevacizumab, ipilimumab, nivolumab, metastasectomy, therapeutic conventional surgery, laboratory biomarker analysis, biopsy
Pembrolizumab With or Without Axitinib for Treatment of Locally Advanced or Metastatic Clear Cell Kidney Cancer in Patients Undergoing Surgery	NCT04370509	Recruiting	Proportion of participants with >2-fold increase in tumor-infiltrating immune cells	Axitinib, pembrolizumab, metastatectomy, cytoreductive nephrectomy

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
