# Peer review of "New Paradigms for Cytoreductive Nephrectomy"

_cancers, 2022, doi:10.3390/cancers14112660_

Round 1

Reviewer 1 Report

The authors should be congratulated for this straightforward clinical review of a very interesting topic as the sense and value of cytoreductive nephrectomy in metastatic clear cell renal cell carcinoma. Only minor concerns need clarification:

  1. When the authors mention high-level publication (lines 234 & 274) it may be more appropriate to say "publication with high level of evidence"
  2. What does "IO" stand for ? Please clarify (line 298)
  3. When the authors mention "early integration of palliative care", what do they exactly refer to? (line 299)

Author Response

Point 1: When the authors mention high-level publication (lines 234 & 274) it may be more appropriate to say "publication with high level of evidence"

Response 1: Edited the verbiage to make it more appropriate. 

Point 2: What does "IO" stand for ? Please clarify (line 298)

Response 2: IO=Immune-oncologic. Thank you for pointing this out. 

Point 3: When the authors mention "early integration of palliative care", what do they exactly refer to? (line 299)

Response 3: Added the following sentence - "Many authors argue that palliative care should be introduced at the time of diagnosis or parallel to curative treatment in cases of life-limiting diseases"

Reviewer 2 Report

This review was reported the utility of cytoreductive nephrectomy in patients with metastatic renal cell carcinoma in the contemporary immunotherapy era. Overall, this paper is well written. The reviewer thinks that this paper have many useful information for readers. The reviewer would like to suggest one critique as follows.

Major revisions

On line 105, the authors reported the results of the CARMENA trial. The reviewer thinks that this study had many problems should be careful about the interpretation of the results. The authors should describe these points.

Author Response

Point 1: "On line 105, the authors reported the results of the CARMENA trial. The reviewer thinks that this study had many problems should be careful about the interpretation of the results. The authors should describe these points."

Response 1: Added the following sentences to further support the notion that the CARMENA trial had many critiques and it should be viewed quite critically - "There was also significant cross-over between the two groups with 15% of the patients in the nephrectomy group not receiving sunitinib and 17% of the patients in the sunitinib only group undergoing a nephrectomy. In addition, the 18.4-month OS reported in the sunitinib group was lower than other previously published reports, again raising questions about patient selection and generalizability."